Journal of Machine Learning Research (2024) 1-11          Submitted 7/24; Revised 9/24; Published 10/24

# *Histopathobiome* – integrating histopathology and microbiome data via multimodal deep learning

**Agata Polejowska**                                    AGATA.POLEJOWSKA@RADBOUDUMC.NL

**Annemarie Boleij**                                    ANNEMARIE.BOLEIJ@RADBOUDUMC.NL

**Francesco Ciompi**                                    FRANCESCO.CIOMPI@RADBOUDUMC.NL

*Department of Pathology, Radboud University Medical Center, Nijmegen, The Netherlands*

**Editor:**

## Abstract

We introduce *Histopathobiome*, a term representing the integration of histopathology and microbiome data to explore tissue-microbe interactions. Using a dataset of colon biopsy whole-slide images paired with microbiota composition samples, we assess the benefits of combining these modalities to distinguish patients with inflammatory bowel disease (IBD) subtype – ulcerative colitis (UC) from non-IBD controls. Initially, we evaluate the unimodal performance of state-of-the-art algorithms using vectors representing bacterial species abundances or histopathology slide-level embeddings. We compare single-modality models with bimodal networks with various fusion strategies. Our results prove that histopathology and microbiome data are complementary in UC classification. By demonstrating improved performance over single-modality approaches, we prove that bimodal deep learning models can be used to learn meaningful and interpretable cross-modal tissue-microbe patterns.

**Keywords:** Computational pathology, microbiome, multimodality, fusion, deep learning, inflammatory bowel disease, ulcerative colitis

## 1 Introduction

We investigate tissue-microbe interactions in inflammatory bowel disease (IBD) through the application of multimodal learning techniques fusing histopathology and microbiota abundance representations. The rationale for combining these two modalities lies in their complementary nature. Histopathology provides detailed insights into tissue morphology and pathological changes, such as ulcers, inflammation, and crypt distortion, which are important in diagnosing and staging diseases like ulcerative colitis (UC) and Crohn's disease (CD), the primary IBD subtypes. However, some features might overlap between them making it more challenging to use only histopathology for definite diagnostics (Kellermann and Riis (2021)). Also, histopathology may not capture the underlying microbial composition and dynamics that can influence these pathological changes. Conversely, microbiome data can reflect the community of microorganisms residing in the tissue and provide information on microbial composition and potential dysbiosis — an imbalance in microbial populations often associated with diseases (Nishida et al. (2018); Wei et al. (2021)). For example, known microbes, such as *Escherichia coli* and *Bacteroides fragilis*, have been implicated in UC pathogenesis (Bruggeling et al. (2023)). Yet, relying solely on microbiome data for diagnosis is challenging due to the dynamic nature of microbial communities, which

can be influenced by various factors and subjected to the quality of preparation methods (Garrett (2019); Hajjo et al. (2022); Schlaberg (2019)).

Thanks to recent advancements in computational pathology and metagenomics sequencing technology, the horizons for artificial intelligence to help understand the complexities behind diseases have been broadened. Computational pathology models are now capable of generating efficient slide-level representations in the form of embeddings, enabling compression of large histology slides into one vector (Xu et al. (2024); Song et al. (2024)). Meanwhile, metagenomics sequencing can provide interpretable microbial profiles (Shi et al. (2022)). However, integrating heterogeneous data, such as histopathology and microbiome data, into a common, learnable representation is still a challenge. Over the years, several fusion techniques and multimodal architectures have been proposed. The basic categories of fusion include early fusion that combines raw data early in the pipeline using simple operations like concatenation, but this may remove structural modality-specifc information. Another popular method is late fusion that uses outputs from separate models trained on different modalities, preserving modality-specific features but potentially missing cross-modal interactions. Recent multimodal frameworks (Hemker et al. (2023), Jaume et al. (2024)) address common drawbacks encountered in basic fusion architectures and demonstrate the potential to efficiently capture cross-modal information while preserving modality-specific features and being able to outperform other baselines.

To our knowledge, our work presents the first application of deep learning to integrate histopathology and microbiome data. The primary aim is to verify whether a deep learning model can depict their complementary nature and thus improve classification of UC patients in remission versus non-IBD patients. We leverage advancements in computational pathology and metagenomics to use a common representation format. We conduct experiments on each modality independently using well-established models. We design and adapt distinct bimodal models by testing different fusion techniques while exploring their abilities to capture cross-modal patterns. The results are concluded not only based on various performance metrics but also by including interpretability measures as the focus is also placed on discovering new tissue-microbe interactions to broaden the understanding of IBD.

## 2 Methods

**Dataset**  The dataset originates from an investigation of the association of the mucosal microbiome, bacterial oncotraits with neoplasia development in UC-patients (Bruggeling et al. (2023)). Biopsies were collected from the ascending and descending colon of 80 UC patients and 35 non-IBD controls. Inclusion criteria for UC patients included: $> 8$ years of disease, left-sided or pancolitis and patients in clinical remission. The ground truth was provided by a gastrointestinal pathologist. The detailed bacterial species composition from clinical tissue samples was derived thanks to isolating the bacterial DNA using an optimized method and shotgun metagenomic sequencing (Bruggeling et al. (2021)). Haematoxylin and Eosin (H&E) stained pathology slides were analyzed together with other techniques. For our experiments, from the original study we selected two modalities: H&E images and relative abundances of bacteria taxa at genus level. Initially, 234 tissue samples were collected, from which we excluded 32 samples due to being out of focus. For image processing, tissue samples were divided into non-overlapping patches of $224 \times 224$ pixels at a spacing of 0.5

micrometers per pixel, resulting in a total of 30644 patches for the entire dataset. Using the extracted patches and their coordinates, we leveraged a foundation model's ability to generate slide-level representations. Specifically, we used a recent Prov-GigaPath model pretrained on a large pathology dataset in which bowel tissue is the second most common tissue type (Xu et al. (2024)). For the microbiome modality, we use raw bacterial composition values at genus level stored in tabular form. Figure 1 provides a high-level overview of the data. We use pairs of microbiota composition from the tissue sample and the corresponding tissue histopathology image from the left and right side of the colon. For each histopathology whole-slide image, we used 768 values in the feature vector as a slide-level representation. For microbiota data, we obtained a vector length of 432 representing relative abundance values for each of 432 bacterial taxa classified at genus level.

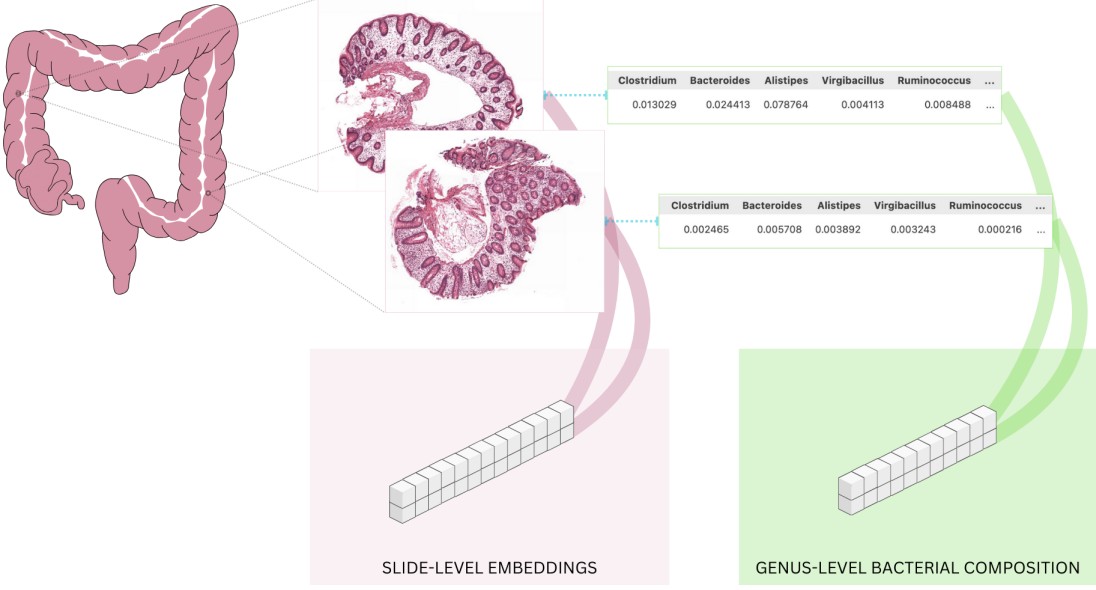

Figure 1: High-level visualization of the dataset preparation from colon biopsies images and microbiota composition at genus level to vector representations.

**Unimodal models** For both, slide-level embeddings and genus-level bacterial composition, we employ deep learning models such as Feature Tokenizer + Transformer (FT-Transformer) (Gorishniy et al. (2021)), Self-attentive Neural Networks (AutoInt) (Song et al. (2019)) and Gated Adaptive Network for Deep Automated Learning of Features (GANDALF) (Joseph and Raj (2022)) within PyTorch Tabular framework (Joseph, 2021). For comparison, we also check the performance of an *ensemble* gradient-based machine learning approach, namely GRANDE (Marton et al. (2023)). From the deep learning approaches, FT-Transformer was chosen for the experiments as it has proven to be a versatile model for a wide range of tabular data problems despite being defined as a simple adaptation of the Transformer architecture to the tabular data domain (Gorishniy et al. (2021)). The AutoInt model is also included in the experiments due to its ability to learn feature interactions in an explicit manner via using an interacting layer which determines the fea-

tures' relevance. GANDALF was selected due to its reported robust feature selection and learning capabilities, excelling in datasets with numerous features or complex feature-target relationships. Also, empirical results demonstrate GANDALF's better performance over a plain Multi-layer Perceptron (MLP) (Joseph and Raj (2022)).

**Bimodal models**  For our use case, we define multimodal, specifically bimodal, models as networks capable of learning cross-modal information from two inputs coming from two different data domains. Despite having transformed the image data into a common format aligned with the microbiota modality, the meaning of the values is different and each modality requires distinct registration and derivation techniques. Firstly, we use the GANDALF architecture on combined histopathology embeddings and microbiota abundances together as parallel input fed into the network. This network was selected as it performed the best among the introduced tabular models showing that it can learn from a data structure with a high number of features. Next, we design and test vanilla fusion models. The early fusion model firstly combines histopathology features and microbiota features via a concatenation operation. The combined input undergoes layer normalization to stabilize the training process. Then, linear layers are used to project the values onto a hidden representation. The late fusion model is built with separate networks to process each input feature independently. The outputs of these two networks are concatenated and fed into a simple fully connected fusion network. Also, we employ more complex fusion techniques. We adapt the Hybrid Early-fusion Attention Learning Network (HEALNet) (Hemker et al. (2023)) for learning from whole-slide image embeddings and microbiota abundance vectors, both defined in a tabular structure for the network input. The next network tested, inspired by SurvPath work (Jaume et al. (2024)), is comprised of a self-normalizing neural network (SNN) block for encoding microbiota vectors and a simple linear projection layer for WSI embeddings. The tokens from those blocks are concatenated and fed into the Nystrom Attention (NA) module. The embeddings obtained after this self-attention approximation are used in the network learning process.

**Training**  The models are trained to classify whether a sample belongs to an individual with confirmed UC or a non-IBD control group. As this is a binary classification, each model network uses Binary Cross Entropy loss during the learning procedure. The length of the training is dependent on the convergence speed and criteria. For this, an early stopping callback is set to stop the learning when the validation loss function does not decrease further after 5 consecutive epochs (patience number equal to 5). The maximum number of epochs is set to 50 with a batch size of 16, a learning rate equal to 0.001 and an Adam optimizer. For training, 64% of the dataset is used.

**Validation and testing**  KFold (k=5) cross-validation is performed. During the validation and testing phase, group ratios are maintained in each split without patient overlap. The validation includes 16% of the dataset. The metrics include: the Area Under the Receiver Operating Characteristic Curve (AUC), Average Precision (AP), and the normalized Matthews Correlation Coefficient (MCC) as it has been reported to address the other metrics' drawbacks (Chicco and Jurman (2023, 2020)). However, as there is also evidence that the MCC alone might not be a suitable measure for an imbalanced dataset (Zhu (2020)), we also compute AP and AUC. The metrics values reported for cross-validation are the mean and standard deviation measured across all folds. For a fixed test set (20% of the

whole data), the best performing model based on the cross-validation mean MCC score is selected, and metrics on the test set for this model are reported.

**Interpretability** To gain insights into tissue-microbe relations, we designed a pipeline to interpret the model's decisions. Based on the conducted experiments, and to be able to look into the unimodal model feature importance assignment process as well as the model decisions that use both modalities, we select the GANDALF model. We use interpretability methods, such as Feature Ablation and GradientShap, implemented in Captum framework (Kokhlikyan et al. (2020)) available for the models via PyTorch Tabular interface (Joseph (2021)). When whole-slide image embedding indices are used as features, we perform patch ablation, checking which patch contributes to the greatest change in the corresponding value in the feature vector. The explanations are run on the whole test set.

## 3 Results and Discussion

Based on Table 1, the most relevant difference in metrics is seen on the test dataset, where the GANDALF model leads. Since it does not outperform other methods on validation data, this suggests GANDALF may benefit from delayed generalization, with its true predictive power emerging on unseen data. Figure 2 indicates that *Absiella*, *Leuconostoc*, *Lactococcus* and *Sporanaerobacter* have the highest attributions for the UC vs. non-IBD classification.

Table 1: Metrics calculated for microbiota abundances input modality.

| Model | Params | Cross-validation | | | Test | | |
|---|---|---|---|---|---|---|---|
| | | AUC | AP | MCC | AUC | AP | MCC |
| AutoInt | 56.1 K | **0.5262 ± 0.0901** | **0.7721 ± 0.0867** | 0.5000 ± 0.0528 | 0.5031 | 0.7884 | 0.5303 |
| FT-Transformer | 300 K | 0.5110 ± 0.0274 | 0.7563 ± 0.1040 | **0.5175 ± 0.0298** | 0.5078 | 0.7929 | 0.5065 |
| GANDALF | 6.7 M | 0.5183 ± 0.1017 | 0.7603 ± 0.1016 | 0.5096 ± 0.0470 | **0.6161** | **0.8317** | **0.5494** |
| GRANDE | NA | 0.4969 ± 0.0054 | 0.7499 ± 0.1049 | 0.4956 ± 0.0076 | 0.5 | 0.7903 | 0.5 |

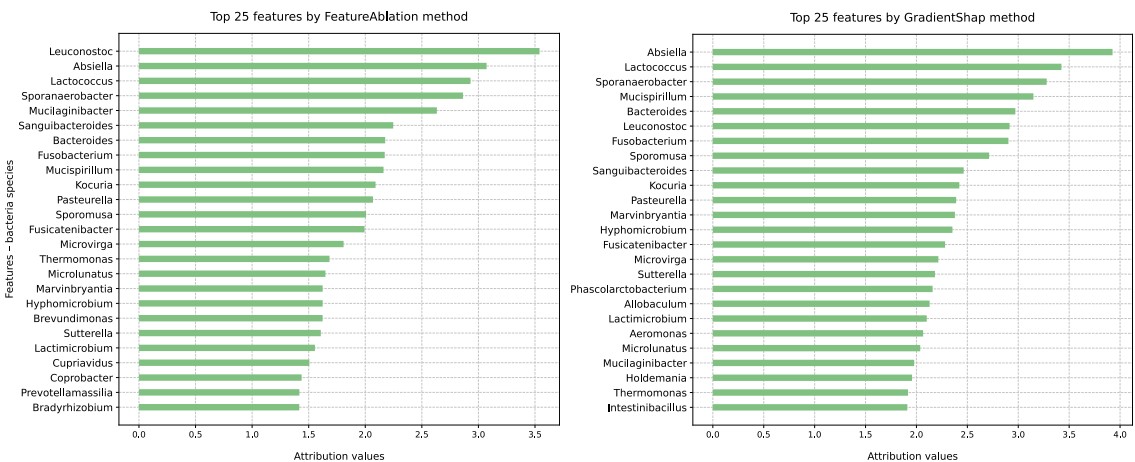

Figure 2: Attribution values for each feature based on two interpretability methods.

Results in Table 2 show that the self-attentive neural network (AutoInt) outperforms other models in this classification task based on whole-slide image embeddings. For congruity, embedding index relevance is computed for the GANDALF model (Figure 3), so that the most important patches for a given embedding index can be verified. We provide visualizations of the most impactful patches (Figure 5) for the most relevant index 621 value for the unimodal model as well as index 399 value since it appears as the feature with one of the highest attribution values in both settings (see Figure 3 and Figure 4).

Table 2: Metrics calculated for histopathology slide-level embeddings input modality.

| Model | Params | Cross-validation | | | Test | | |
|---|---|---|---|---|---|---|---|
| | | AUC | AP | MCC | AUC | AP | MCC |
| AutoInt | 89.0 K | $0.5537 \pm 0.0521$ | $\mathbf{0.7363 \pm 0.0995}$ | $\mathbf{0.5588 \pm 0.0582}$ | **0.6259** | **0.7837** | **0.6358** |
| FT-Transformer | 323 K | $\mathbf{0.5548 \pm 0.0546}$ | $0.7358 \pm 0.1060$ | $0.5725 \pm 0.0728$ | 0.5814 | 0.7634 | 0.6326 |
| GANDALF | 21.3 M | $0.5247 \pm 0.0940$ | $0.7308 \pm 0.1009$ | $0.5445 \pm 0.0830$ | 0.5444 | 0.7477 | 0.5577 |
| GRANDE | NA | $0.4921 \pm 0.0559$ | $0.7112 \pm 0.1105$ | $0.4971 \pm 0.0954$ | 0.5315 | 0.7424 | 0.5618 |

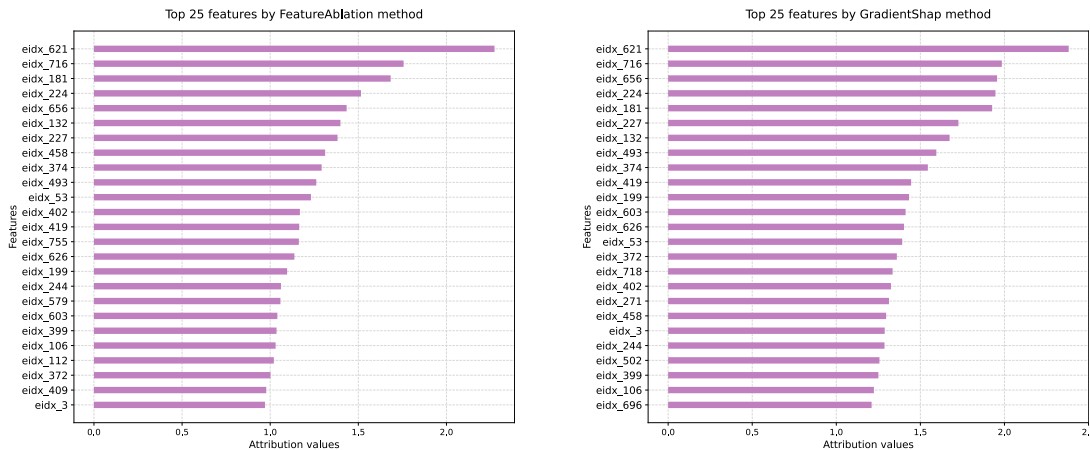

Figure 3: Attribution values for each feature computed using two interpretability methods.

According to the obtained results, particularly the AP score on test data, microbiome-only data shows some informative value for classifying IBD patients. However, when considering performance across all models and metrics, histopathology slides embeddings tend to outperform microbiome data overall. This suggests that the superiority of one modality is not definitive, highlighting the challenge of using a single modality for optimal performance. Therefore, performance gains may be achievable by employing fusion techniques to merge both modalities and capture interconnected patterns.

For multimodal performance (Table 3), the self-normalizing network together with self-attention approximation module gives the best results on the test dataset given AUC and MCC metrics. Simpler approaches, like Early Fusion and Late Fusion, show even better performance, according to AP values, compared to more complex ones. Also, the GANDALF model benefits from learning on merged representations achieving improved results

on the test set based on AUC and MCC values in contrast to when used for single-modality experiments. In Figure 4, it can be seen that microbiota abundances largely dominate the model's decision-making process. Index 399 and 717 of whole-slide image embeddings appear twice in both interpretability methods. Values in those indices are mostly impacted by the patches shown in Figure 5. It can be observed that more focus is placed on colon crypts in the case of non-IBD tissue samples. This is reasonable as in ulcerative colitis, the crypts might be distorted. Additionally, some patches capture mainly cells which are an essential feature to be inspected as a high number of inflammatory cells is attributed to UC (Rubio and Schmidt (2018)).

Table 3: Metrics calculated for different multimodal algorithms.

| Model | Params | Cross-validation | | | Test | | |
|---|---|---|---|---|---|---|---|
| | | AUC | AP | MCC | AUC | AP | MCC |
| GANDALF* | 51.9 M | $0.6412 \pm 0.1438$ | $0.7962 \pm 0.1045$ | $0.5519 \pm 0.0604$ | 0.6733 | 0.8133 | 0.5777 |
| Early Fusion | 617 K | $0.6942 \pm 0.0997$ | $0.8488 \pm 0.0999$ | $0.5606 \pm 0.0471$ | 0.6578 | 0.8572 | 0.6451 |
| Late Fusion | 1.1 M | $0.7345 \pm 0.0431$ | $\mathbf{0.8785 \pm 0.0552}$ | $0.6327 \pm 0.1148$ | 0.68 | **0.8689** | 0.5915 |
| NA + SNN | 1.8 M | $\mathbf{0.7627 \pm 0.0790}$ | $0.7513 \pm 0.0828$ | $\mathbf{0.7472 \pm 0.0844}$ | **0.8015** | 0.8357 | **0.7353** |
| HEALNet | 32.2 M | $0.7472 \pm 0.0844$ | $0.6949 \pm 0.0925$ | $\mathbf{0.7472 \pm 0.0844}$ | 0.7353 | 0.7151 | **0.7353** |

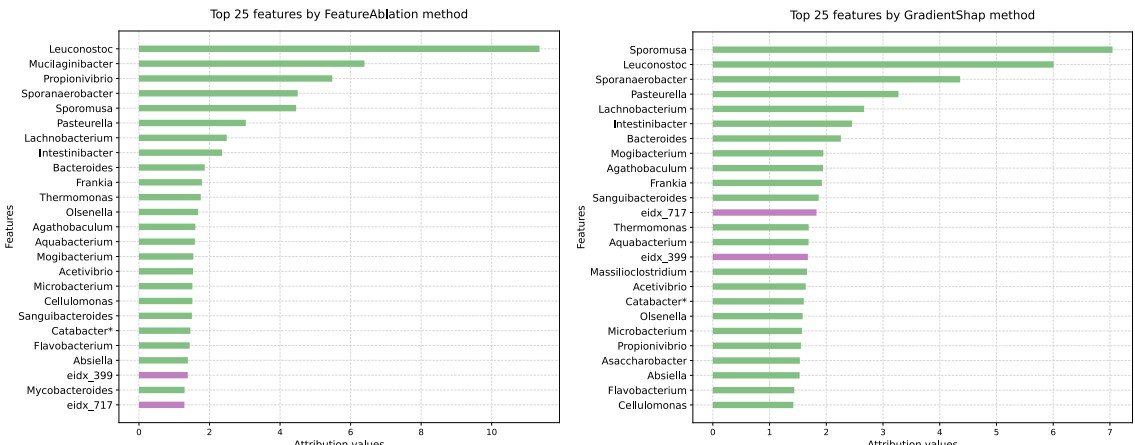

Figure 4: Attribution values for each feature based on two interpretability methods.

When compared to input based solely on microbiota (Figure 2), *Leuconostoc* bacteria continue to be ranked at the top (Figure 4). Interestingly, a recent study has reported that *Leuconostoc* species, which are found in kimchi (Moon et al. (2023)), are capable of attenuating the inflammatory responses associated with colitis. In fact, the control group has, on average, higher levels of *Leuconostoc* than the colitis group in our dataset.

Furthermore, when adding histopathology data, some bacteria like *Intestinibacter*, *Lachnobacterium* and *Agathobaculum* become more important, while others like *Marvinbryantia* and *Kocuria* no longer appear in the top rankings. The increased importance of *Intestinibacter* is promising as a new finding, congruent with the reported association of this genus with lower genetic risk for autoimmunity in the gut (Russell et al. (2019)).

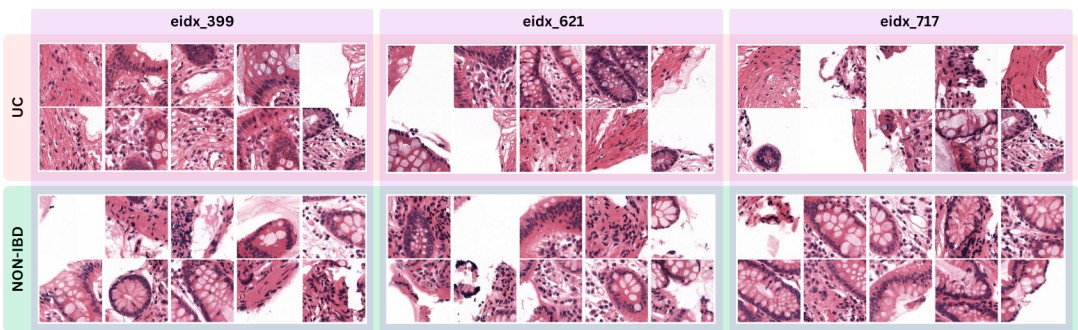

Figure 5: Most influential patches from UC and non-IBD patient on embeddings' values based on embedding's indices with the highest attribution value.

Overall, the observations presented prove that a multimodal approach can reveal scientifically plausible insights into how specific bacteria can impact tissue and thus host health.

## 4 Limitations

Further potential enhancement of model performance might be achieved through the implementation of alternative architectures and experimental set-ups. As model performance improves, the reliability of interpretability method outputs is expected to increase. However, the selection of interpretability methods requires careful consideration. For instance, tests with KernelShap demonstrated a higher prevalence of histopathology features in the attribution plots. Further, the exploration of other pretrained models for generating whole-slide image embeddings warrants further investigation. Next, the relevance of patches for histopathology and microbial abundance needs to be confirmed with experts and larger validation cohorts to confirm the biology and performance of the models.

## 5 Conclusions

This study contributes the foundation for future development of the new *Histopathobiome* learning and exploration application. The experimental results suggest that the results obtained might reflect the inherent complexity of the dataset or UC in general. Nevertheless, combining different features using bimodal models can lead to better results compared to single-modality-based algorithms, thus proving *Histopathobiome* potential. Furthermore, we demonstrated the exploration of tissue-microbe patterns, using interpretability techniques, and confronted the outcomes with the literature. This application of artificial intelligence shows promise in uncovering previously unknown associations and demonstrates the benefits of combining histopathology with microbiome data. This approach may extend to other diseases and conditions as understanding the biological environment and the coexistence with bacteria can lead to exploring their potential to treat or prevent various diseases as we deepen our understanding of their impact with the help of deep learning.

**Acknowledgments and Disclosure of Funding**

This work was supported by the HEREDITARY Project, as part of the European Union's Horizon Europe research and innovation programme under grant agreement No GA 101137074.

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
