# OpenReview forum: "Histopathobiome – integrating histopathology and microbiome data via multimodal deep learning"
_MICCAI.org/2024/Workshop/COMPAYL — COMPAYL 2024_

### Official Review · Reviewer_7PTk · 2024-07-08
**Original and timely paper**

**Custom Rating:** 5
**Confidence:** 5

**Review:**

This work is a nice example of bimodal data integration, using histopathology and microbiome data (“histopathobiome”) to tackle the issue of classifying patients with ulcerative colitis from non-IBC patients. One question arises with regards to the ground truth used for these images- where does it come from? Also, the current classification of UC depends on 3 features: DCA  (distribution, chronicity and activity), where a score of 0 for each implies a normal biopsy. How do the models compare to the DCA score? One of the features (D) is not actually related to histomorphology per se, but rather to the area of the biopsy which is affected. This feature could of course not turn up in the patch-wise approach.

---

### Official Review · Reviewer_unpZ · 2024-07-10
**Review of Histopathobiome**

**Custom Rating:** 4
**Confidence:** 4

**Review:**

The authors explore the impact of combining tissue based (from histopathology slides) and microbiome data for Inflammatory Bowel Disease (IBD) prediction. They use a foundational model (FM) to extract slide-level representations, whereas for microbiome data they use a vector representation in which each dimension corresponds to a bacteria, and the values represent their relative abundance.

Pros:
- The work proposes a novel integration of two modalities: histopathology and microbiome.
- The authors conduct a cross-validation analysis and explore different methods to prove their claim: the interaction between tissue and microbiome leads to better understanding of IBD.
- They also conduct an interpretability analysis for each experiment.
- Their results match with existing literature for microbiota data.
- Experimental choices and decisions are well justified and explained across the text.
- The manuscript is well written and easy to follow.

Cons/Comments:
- How was the data partitioned? Train + test were divided in a stratified manner, and then train is used for 5-fold cross-validation? Given the very small size of the dataset, as well as its imbalance, it would have been interesting to see an averaged performance across different train-test splits, to see the variability of the results for different data partitions. For instance, it's surprising that test metrics in Table 1 and Table 2 are significantly larger than cv ones for some methods.
- The multi-modality is addressed by modality independent encoders (actually only for tissue) and then combined. It would be interesting to explore how to combine these modalities at earlier stages.

---

### Official Review · Reviewer_inp7 · 2024-07-12
**Exploratory work on multi-modal learning combining WSIs and microbiome data**

**Custom Rating:** 4
**Confidence:** 4

**Review:**

**Overview**

In this paper, the authors explore combining imaging data from Whole Slide Images (WSIs) with microbiome data from the colon to predict inflammatory bowel disease. The simultaneous utilization of these data is the main novelty of the paper. The authors explore different approaches, such as late and early fusion. Results show the benefits of combining multi-modal data. Furthermore, the authors attempt to interpret the most relevant features of the task. The methodology is simple, serving well the purpose of this exploratory work in an unexplored setting.

**Pros**

1. The text is well-written and easy to follow.
2. The motivation for combining imaging data with microbiome data is sound.
3. Exploring this combination of data is novel.
4. Using the multi-modal approach significantly improves results, thus demonstrating its usefulness.

**Cons**

1. The dataset is relatively small, with 202 tissue samples (after excluding 32 due to out-of-focus).
2. While combining modalities helps, the results of unimodal approaches are intriguing. Some of the entries in Tables 1 and 2 show AUROC just around or slightly above 0.5, suggesting the classifier performs similarly or just a little better than a random classifier.


**Further comments**

Regarding the results:

1. In Table 1, the results of the validation and test set are similar for most methods. The exception is GANDALF, where the performance of the test set significantly improves. This is different from the other methods, and somewhat unexpected since validation is used to tune the models. Could the authors elaborate on what may be the reasons?
2. In Figure 5 the authors show influential patches for the predictions. It would be interesting if the authors could comment on the content of the patches and if the patterns make sense for the task at hand.
3. As mentioned in “Cons” the AUROC is relatively low for unimodal approaches but increases significantly with the multi-modal approaches.

    a. In the unimodal methods (Table 1 and 2) we observe AUROC mostly in the range [0.5, 0.6] and some around 0.5, being close to a random classifier. It would be interesting if the authors could comment on this aspect.

    b. In Table 3, however, several methods exhibit AUROC > 0.7, and the best is 0.8. This implies that combining modalities is effective.

    c. However, it was very challenging for unimodal approaches. Perhaps, the interaction of the data unlocks key aspects for better predictions. Shedding light on this could be interesting in future research.

---

### Decision · Program_Chairs · 2024-07-16

Accept